# Plasma Proteomic Profiling Reveals the Regulatory Factors of Milk Protein Synthesis in Holstein Cows

**DOI:** 10.3390/biology11081239

**Published:** 2022-08-19

**Authors:** Xinling Wang, Jie Xu, Zhaoyu Han

**Affiliations:** College of Animal Science and Technology, Nanjing Agricultural University, Nanjing 210095, China

**Keywords:** milk protein, plasma proteomic, bovine mammary epithelial cells, insulin-like growth factor 1

## Abstract

**Simple Summary:**

We aimed to determine the plasma bio-markers of cows with high long-term milk protein concentration and investigate the mechanism of plasma proteins in milk protein synthesis. Some plasma proteins are considered to be related to milk protein synthesis. However, the characteristics of these proteins with different long-term milk protein concentrations are not fully elucidated. In this study, we collected milk protein concentration data of Holstein cows for 10 months on a commercial farm. Three groups of cows (*n* = 10 per group) with low, medium, and high milk protein concentrations were selected. We found that cows with high milk protein concentration had higher glucose, insulin-like growth factor 1, prolactin, insulin, and growth hormone concentrations than cows with low milk protein concentration. A total of 91 differentially expressed proteins were identified between cows with high and low milk protein concentrations by plasma proteomic. Furthermore, β-casein level and mammalian rapamycin target protein pathway in bovine mammary epithelial cells were enhanced by insulin-like growth factor 1 treatment. Our findings revealed that the differences in plasma proteins in cows with different milk protein concentrations and determined that β-casein synthesis was increased via the mammalian rapamycin target protein pathway following IGF-1 treatment in vitro.

**Abstract:**

Milk protein concentrations in dairy cows are considered to be related to some plasma biomolecules. However, the characteristics of plasma biomolecules in dairy cows with different long-term milk protein concentrations are not fully elucidated. This study was conducted to understand the mechanism of plasma proteins in milk protein synthesis by the comparative analysis of the plasma proteomics of cows with different milk protein concentrations. Three groups of Holstein cows (per group = 10) with low (LMP), medium (MMP), and high long-term milk protein concentrations (HMP) were selected for the experiment to determine plasma hormones, biochemical parameters, and proteome. We found that HMP cows had higher concentrations of plasma insulin-like growth factor 1 (IGF-1), glucose, prolactin, insulin, and growth hormone than LMP cows. Additionally, plasma proteomic identified 91 differential proteins, including IGF-1 between the LMP and HMP groups, and the mTOR pathway was enriched. In vitro, IGF-1 treatment increased β-casein expression and simultaneously activated S6K1 and mTOR phosphorylation in bovine mammary epithelial cells. Taken together, these data demonstrate the differences in plasma hormones, biochemical parameters, and proteome of cows with different milk protein concentrations and indicate that IGF-1 enhanced milk protein synthesis, associated with activation of the mTOR signaling pathway.

## 1. Introduction

Milk protein concentration is generally used as an important indicator of milk quality due to containing an abundance of amino acids that are easily absorbed and utilized by the human body. Increasing the protein concentration in milk has always been one of the most urgent problems in animal husbandry. Milk protein is one of the most variable components of milk and is affected by many factors, such as dairy cow breed, nutrition, and parity. Recently, we reported that long-term differences in milk protein concentrations in Holstein cows could also be observed under common feeding conditions [1]. Thus, it is necessary to investigate the reason for the difference in milk protein concentrations in cows under the same feeding conditions.

Some biomolecules and hormones can be transported in the blood to the mammary gland, which is the main synthetic site of milk protein and further affects the synthesis of milk protein. A previous study showed that intravenous injection of glucose (Glc) into previously fasted lactating cows increased milk protein production [2]. In addition, the levels of some hormones and biochemical substances could effectively reflect the physiological status and, most notably, could reflect the lactation performance of dairy cows. Some components of the blood in the tail artery of dairy cows are similar to those in the external pudendal artery, both of which originate from the same branch of the artery [3]. Due to the difficulty of collecting blood from the external pudendal artery, tail artery blood is often used to represent the blood flowing into the mammary gland in previous studies [4,5]. Moreover, at the molecular level, milk protein synthesis is mainly regulated by the mammalian rapamycin target protein (mTOR) signaling pathway, which is crucial for the translation process of milk protein synthesis. Previous studies have shown that the mTOR signaling pathway is regulated by a variety of hormones and nutrients. Burgos et al. suggested that some nutrients and hormones such as prolactin (PRL), cortisol (CL), and insulin (INS) may effectively activate the mTOR pathway, thereby affecting protein synthesis in isolated mammary acini [6]. An in vitro study showed that IGF-1 stimulates the mTOR signaling pathway in MAC-T cells [7]. A previous study suggested that the increase in milk protein yield by injecting growth hormone (GH) into lactating cows may be mediated by the initiation and prolongation (PRL) of mRNA translation via the mTOR signaling pathway [2]. Thus, investigating the differences in specific plasma hormones and biochemical indices of dairy cows with different milk protein concentrations may help in understanding the regulation role of these plasma biomolecules in milk protein synthesis.

Proteomic is one of the important fields of study that describes the composition of cellular proteins and the functional connection between these proteins to achieve a deep understanding of the mechanisms of biological processes. The genome remains relatively stable in the process of individual development; however, proteins will constantly change [8]. Therefore, protein activity was the final functional product of these genes. As proteomic technology has developed in recent years, data-independent acquisition (DIA) has the characteristics of good discrimination and reproducibility. Plasma was used in proteomic research due to accessibility, and can comprehensively interpret the molecular regulation mechanism related to a certain trait in terms of plasma proteins and contribute to the exploration of related biomarkers. Herosimczyk et al. detected age-dependent and diet-dependent plasma proteome changes during the early postnatal period using proteomics technology [9]. A previous study assessed the effect of lipid supplementation on the proteomic profile of mammary gland tissue in lactating goats using proteomics technology and found that a high-lipid diet interfered with the expression of many proteins [10]. Additionally, a previous study detected 3 and 41 unique proteins using proteomics technology in plasma exosomes of high-fertility heifers and low-fertility heifers, respectively, and some protein differences were also observed, which contributed to the identification of potential markers of dairy cow fertility [11]. Nevertheless, in the available literature, there is a lack of information concerning plasma proteomic changes in cows with different milk protein concentrations.

Thus, we aimed to investigate the differences in plasma hormones, biochemical indices, and proteomes of cows with different long-term milk protein concentrations. We aimed to determine potential plasma biomarkers of high milk protein concentrations in Holstein cows under the same feeding conditions and provide new ideas for studying the molecular mechanism of milk protein synthesis.

## 2. Materials and Methods

### 2.1. Animals and Experimental Design

All animal procedures complied with the guidelines of the Experimental Animal Welfare and Ethics Committee of the Nanjing Agricultural University.

The experimental treatment procedure has been previously described [1]. Briefly, milk protein concentrations of cows over 10 months (10 points) were collected from dairy herd improvement (DHI) data from a commercial farm. A total of 30 cows with a similar lactation period, milk yield, and parity were selected and divided into three groups (*n* = 10 per group): cows with low milk protein concentration (LMP) group (milk protein concentration < 3.1%), medium milk protein concentration (MMP) group (3.1% ≤ milk protein concentration < 3.4%), and high milk protein concentration (HMP) group (milk protein concentration ≥ 3.4%). Notably, the milk protein concentrations of these experimental dairy cows throughout the 10 months (10 points) met the above grouping requirements, and the cows were in the late lactation stage (lactation days = 313 ± 5.86, mean ± SEM) when sampled. The cows were managed in a tie-stall configuration a month before sampling, fed with the same total mixed ration, and given free access to water.

### 2.2. Plasma Sampling

Blood samples of the 30 cows were obtained on the same day, and sampled only once. Caudal artery blood was collected in anticoagulant vacuum tubes with EDTA before early morning feeding. The blood samples were centrifuged at 3000× *g* at 4 °C for 15 min, and then the plasma was separated and stored at −80 °C until further processing.

### 2.3. Biochemical Index Measurement

Plasma biochemical indices were measured using the enzymatic colorimetric method according to previously described methods [12]. Briefly, plasma sample was transferred to colorimetric cylinder, and the concentration of the biochemical index was obtained using an AU2700 autoanalyzer (Olympus, Tokyo, Japan).

### 2.4. Hormone Concentration Assay

Hormone concentrations in plasma were detected using the double antibody sandwich ELISA kits (Fangcheng, Beijing, China) according to the manufacturer’s protocol, and the absorbance at 450 nm was measured using a spectrophotometer (SHIMADZU, UV-2450). The concentrations of insulin (INS), growth hormone (GH), cortisol (CL), prolactin (PRL), and insulin-like growth factor 1 (IGF-1) were finally determined by comparing the OD value of the samples to the standard curve using Microsoft Excel.

### 2.5. Plasma Proteomic Analysis

Three of the ten cows in each group were randomly selected for plasma proteome analysis (*n* = 3 per group). The plasma proteome was performed using data-independent acquisition (DIA) technology as previously described [13]. Plasma proteins were extracted as previously described [13]. After enzymolysis of the protein, the samples were redissolved in buffer A (buffer A:20 mM aqueous ammonium formate) and then connected to a reversed-phase column (XBridge C18 column, 4.6 × 250 mm, 5 μm, Waters Corporation, Milford, MA, USA) using an Ultimate 3000 system (Thermo Fisher Scientific, Waltham, MA, USA) for high-pH reversed-phase separation. The reversed-phase column was equilibrated for 15 min under initial conditions. The flow rate and column temperature were maintained at 1 mL/min and 30 °C, respectively.

Samples were redissolved in solvent A (a: 0.1% formic acid aqueous solution) after desalting and freeze drying and then analyzed by LC-MS/MS equipped with an online EASY-Spray nano-electrospray ion source (Thermo Scientific, Waltham, MA, USA). The entire system was an Orbitrap Fusion Lumos Mass Spectrometer (Thermo Fisher Scientific, Waltham, MA, USA) connected in series with an easy NLC 1200 system. The loading amount was 3 μL, and the sample was separated in a 120 min gradient: 5% B to 35% B (B: 0.1% formic acid in acetonitrile). The column flow was controlled at 200 nL/min, and the electrospray voltage was 2 kV.

The Orbitrap Fusion Lumos mass spectrometer was operated in the data-dependent collection mode and automatically switched between MS and MS/MS. The mass spectrum parameters were as follows: (1) MS: scanning range (m/z) = 350–1500, resolution = 120,000, automatic gain control target = 4 × 10^5^, maximum injection time = 50 ms, dynamic exclusion time = 30 s; (2) HCD-MS/MS: resolution = 15,000, automatic gain control target = 5 × 10^4^, maximum injection time = 35 ms, collision energy = 32.

Original data were analyzed, and databases were searched using Spectronaut X (Biognosys AG) and UniProt. The false positive rates (FDR) of both parent ion and peptide levels were set to 1%. Then, 0.1% formic acid aqueous solution was added to the sample to obtain the suspension. The mixture of the sample and 10 × IRT peptides was separated by nano LC after mixing and analyzed by online electrospray mass spectrometry. The experimental system was an EASY-nLC 1200 nano LC system (Thermo Scientific, Waltham, MA, USA) coupled online to an Orbitrap Fusion Lumos mass spectrometer (Thermo Fisher Scientific, Waltham, MA, USA). The mass spectrum parameters were as follows: (1) MS: scanning range (m/z) = 350–1500, resolution = 120,000, automatic gain control target = 4 × 10^6^, maximum injection time = 50 ms; (2) HCD-MS/MS: resolution = 30,000, automatic gain control target = 1 × 10^5^, collision energy = 32, energy increase = 5%. Analyses of principal component analysis (PCA), Kyoto Encyclopedia of Genes and Genomes (KEGG), and differentially expressed protein were performed using the Omicsmart web server (https://www.omicsmart.com/, accessed on 7 August 2021).

### 2.6. Bovine Mammary Epithelial Cells Isolation and Culture

Bovine mammary epithelial cells were isolated from fresh bovine mammary tissue collected from lactating cows according to the sensitivity of bovine mammary epithelial cells (BMECs) and fibroblasts to trypsin, as described in our previous study [14].

### 2.7. Cell Viability Assay

To evaluate the effect of IGF-1 concentration on cell viability, BMECs were first cultured in 96-well culture plates and then were treated with different concentrations (0, 50, 100, 150, and 200 ng/mL) of IGF-1 (Prospec, Rehovot, Israel) for 12 h. Cells were mixed with cell counting kit-8 (CCK-8; Shanghai Biyuntian, Shanghai, China) for 1.5 h after cell density reached approximately 70%. Finally, absorbance was measured at 450 nm using a Multiskan Go microplate reader (Thermo Fisher, Waltham, MA, USA).

### 2.8. Western Blotting Assay

Western blotting was performed as previously described [14]. Briefly, the total protein was extracted using RIPA lysis buffer (Solarbio, Beijing, China) and phenylmethylsulfonyl fluoride (PMSF, Solarbio, Beijing, China), and then the samples were treated with SDS-PAGE protein loading buffer (Biosharp, Beijing, China) at 100 °C for 10 min. The denatured protein samples were separated on 8% and 12% SDS-PAGE gels (Genscript, Nanjing, China) according to the manufacturer’s instructions. Proteins were electrophoretically transferred to polyvinylidene fluoride (PVDF, Bio-RAD Company, Hercules, CA, USA). The PVDF membrane was probed with primary antibodies at 4 °C overnight and then incubated with secondary antibodies conjugated to horseradish peroxidase goat antirabbit IgG (1: 5000, Solarbio, Beijing, China) in Tris-buffered saline with tween (TBST) at room temperature for 1.5 h. Finally, chemiluminescence imaging of the proteins was visualized using an enhanced chemiluminescence reagent (Biosharp, Beijing, China) and a chemiluminescence gel imaging system (ChemiDoc MP; Bio-Rad, Hercules, CA, USA). Image J 1.8.0 software was used to quantify the densitometry readings of the bands. The primary antibodies used in our study were as follows: GAPDH (1: 2000, Proteintech, Chicago, IL, USA), β-casein (1: 500, Bioss, Beijing, China), IGF-1 (1: 1000, Biorbyt, Wuhan, China), mTOR, p-mTOR (Ser2448) S6K1, and p-S6K1 (Thr389) (1: 1000, Cell Signaling Technology, Beverly, MA, USA). 

### 2.9. Statistical Analyses

SPSS version 20.0 (SPSS Inc., Chicago, IL, USA) and GraphPad Prism version 7.00 (GraphPad Software, San Diego, CA, USA) were used for data analysis and graph generation, respectively. Data differences were assessed using one-way analysis of variance (ANOVA) and the least significant difference (LSD) test, and the data were expressed as the mean and standard error of the mean (SEM). *p* < 0.05 was considered statistically significant.

## 3. Results

### 3.1. Plasma Biochemical Parameters and Hormones Concentrations

To assess the differences in blood biochemical parameters and hormones in dairy cows with different milk protein concentrations, we measured the concentrations of some biochemical indices and plasma hormones among the three groups. As shown in Table 1, the concentrations of aspartate aminotransferase (AST) and high-density lipoprotein (HDL) were significantly higher in the LMP group than those in the other two groups (*p* < 0.05). Alanine aminotransferase (ALT) concentration in HMP and LMP cows and globulin concentration in HMP cows were higher than those in the MMP groups (*p* < 0.05). Glc concentration was significantly higher in the HMP group than in the LMP group (*p* < 0.05), while total cholesterol (TCHO) and low-density lipoprotein (LDL) concentrations were lower in the HMP group than in the LMP group (*p* < 0.05). The total protein and globulin concentrations were higher in the HMP group than in the MMP group (*p* < 0.05).

Additionally, as shown in Figure 1, the concentrations of INS (HMP vs. MMP, *p* = 0.031; HMP vs. LMP, *p* = 0.047), PRL (HMP vs. MMP, *p* = 0.031; HMP vs. LMP, *p* < 0.001), and GH (HMP vs. MMP, *p* = 0.008; HMP vs. LMP, *p* < 0.001) in the HMP group were significantly higher than those in the other two groups. Furthermore, HMP cows had a higher IGF-1 concentration than LMP cows (*p* = 0.001), whereas the cortisol (CL) concentration showed a higher concentration in the LMP group than that in the MMP group (*p* = 0.008).

### 3.2. Plasma Proteome

DIA technology was performed to identify plasma protein differences in cows with different milk protein concentrations. A total of 1049 proteins were detected, of which principal component analysis (PCA) showed differences among the three groups (Figure 2A). This study detected 22, 44, and 43 unique proteins using proteomics technology in the plasma of HMP, MMP, and LMP cows, respectively (Figure 2B). The results showed that the MMP group had 22 upregulated and 18 downregulated proteins compared to HMP group, and 43 upregulated and 16 downregulated proteins were observed in the MMP compared to LMP group, respectively (*p* < 0.05). Moreover, 81 upregulated and 10 downregulated proteins were identified in the HMP group compared with the LMP group (*p* < 0.05) (Figure 2C).

Differentially expressed proteins were screened using the filtering conditions of FC ≥ 1.5 and *p* < 0.05. Table 2 lists the proteins that showed significant differences between every two groups among the three groups. Furthermore, as shown in Figure 3, the differentially expressed proteins were subjected to functional enrichment using KEGG analysis (Figure 3). Notably, the AMPK signaling pathway, mTOR signaling pathway, PI3K-Akt signaling pathway, MAPK signaling pathway, and RNA transport pathway were significantly enriched between the LMP and HMP groups (*p* < 0.05) (Figure 3A). Lysine degradation, galactose metabolism, carbohydrate digestion and absorption, and starch and sucrose metabolism were significantly enriched between the MMP and HMP groups (*p* < 0.05) (Figure 3B), and the PI3K-Akt signaling pathway, NF-kappa B signaling pathway, and calcium signaling pathway were significantly enriched between the MMP and LMP groups (*p* < 0.05) (Figure 3C). To confirm the differential expression of plasma proteins, IGF-1 in plasma was analyzed by Western blotting (Figure 4 and Appendix A), which firmly supported our proteomic data.

### 3.3. Effect of IGF-1 Treatment on the β-Casein and mTOR Signaling Pathway

The effect of IGF-1 at different concentrations (0, 50, 100, 150, and 200 ng/mL) on BMEC viability was measured using CCK8, and we found that cell viability was not inhibited by IGF-1, but was significantly increased at a concentration of 200 ng/mL (Figure 5A). The results showed that treatment with 200 ng/mL IGF-1 significantly increased β-casein levels in BMEC, accompanied by increased phosphorylation of mTOR and S6K1 (Figure 5B–E and Appendix A).

## 4. Discussion

To obtain a comprehensive overview of the plasma protein differences in cows with different long-term milk protein concentrations, the concentrations of biochemical parameters, hormones, and proteomes in plasma were analyzed in this experiment. In this study, we found that the concentrations of ASP, HDL, LDL, and TCHO were significantly higher in the LMP group than in the HMP group, whereas Glc and IGF-1 concentrations were significantly higher in the HMP group than in the LMP group. Additionally, the concentrations of INS, PRL, and GH in the HMP group were significantly higher than those in the other two groups. Plasma proteomics showed 22, 44, and 43 unique proteins in the plasma of HMP, MMP, and LMP cows, respectively, and 91 differentially expressed proteins were identified between the LMP and HMP groups, similar to IGF-1. Furthermore, we demonstrated that IGF-1 treatment potentially increased the milk protein synthesis capacity of BMECs in vitro by increasing the mTOR signaling pathway.

By comparing the proteome among cows with different milk protein concentrations, we found that the sort from high to low according to adiponectin (ADIPOQ) levels were HMP, MMP, and LMP groups, respectively, and the pairwise comparison between groups was statistically significant (*p* < 0.05). ADIPOQ is an endogenous bioactive protein secreted by white adipose tissue. Previous studies have proposed that ADIPOQ has a variety of biological functions, including improving animal reproductive performance, modulating the inflammatory response, improving glucose lipid metabolism disorders, and relieving insulin resistance [15,16,17,18,19,20,21]. Moreover, a previous study suggested that ADIPOQ is a new candidate gene associated with litter size in the Awassi sheep [22]. Therefore, the difference in plasma ADIPOQ levels observed among the three groups in our study could be a potential biomarker protein associated with the milk protein concentration of Holstein cows under the same feeding conditions. Additionally, the level of plasma fibronectin 1 (FN1) was significantly higher in the LMP group than in the other two groups. FN1 is a glycoprotein involved in cell adhesion and migration processes, and little is known about the biological function of FN1 in the mammary glands of cows. A previous study showed that FN1 mRNA expression in renal cell cancer was higher than that in normal renal tissue [23]. FN1 concentration seemed to be closely related to the low milk protein phenotype observed in our study. Therefore, our data on ADIPOQ and FN1 encourage further investigation to determine their precise roles in milk protein synthesis.

Milk protein synthesis is an energy-intensive process. Previous studies have reported that plasma Glc and TCHO are useful indicators of energy status, and Glc concentration decreases when energy intake is insufficient [24,25]. In our study, the concentration of plasma Glc was significantly higher in the HMP group than in the LMP group, indicating that the energy intake ability of HMP cows was significantly higher than that of LMP cows. Toerien et al. (2010) found that the milk protein yield of lactating cows that were feed-deprived was increased 27% by Glc jugular infusion for 9 h [26]. An in vitro study also demonstrated that Glc deficiency inhibited casein synthesis through the JAK2/STAT5 and AMPK/mTOR signaling pathways in mammary epithelial cells of dairy cows [27]. Notably, KEGG functional enrichment analysis of our proteomic analysis showed that the AMPK and mTOR pathways were significantly enriched between HMP and LMP cows. The AMPK and mTOR pathways are closely interrelated with the variation in energy in cells, which can be activated and inhibited, respectively, with a decreased energy level. These results indicated that higher plasma Glc concentrations suppressed AMPK pathway activation and activated the mTOR pathway in the HMP group compared with the LMP group. Therefore, higher plasma Glc concentrations may partially explain the contributing factor of the high milk protein concentration in our study. Plasma TCHO levels are related to the process of lipid mobilization [28]. Normally, plasma Glc is preferentially used to supply energy for lactation in dairy cows. Thus, the plasma TCHO concentration of cows in the LMP group was significantly higher than that of cows in the HMP group, which may be due to lipid mobilization caused by insufficient Glc supply in the LMP group. Additionally, TCHO is also involved in the biosynthesis of steroid hormones [29]. Not surprisingly, as a steroid hormone detected in our study, the plasma CL concentration was higher in the LMP group. A previous study pointed out that high AST and ALT levels were positively correlated with the risk of liver disease [30]. The plasma AST concentrations observed in cows with high and low residual feed intake were 88.7 and 91.4 U/L, respectively [31], which were far lower than the 109.8 U/L of AST in the LMP group in our study. Thus, higher concentrations of AST and ALT in LMP cows may indicate a higher risk of disease.

PRL is a polypeptide hormone synthesized and secreted from the anterior pituitary gland. Studies on cows and goats have proposed that milk protein levels increase because of an increase in prolactin [32,33,34,35]. INS is a unique hormone with hypoglycemic activity in the body. It can promote the conversion of glucose to protein by promoting the synthesis of liver and muscle glycogen. Zhang et al. reported that the concentrations of INS and Glc in cows with sufficient prenatal energy supply were higher than those in cows with insufficient energy supply [36], which is consistent with our results. Moreover, a study has also shown that the INS signal affected milk protein production in dairy cows through direct or indirect pathways [37]. INS not only regulates the synthesis of milk protein in synergy with PRL but also individually promotes the expression of genes related to milk protein synthesis [38]. Therefore, high levels of INS and PRL may promote milk protein synthesis, which is consistent with our results that the plasma INS and PRL levels of cows in the high milk protein group were significantly higher than those in the other two groups.

Similarly, plasma GH concentrations in the HMP group were much higher than those in the other two groups in our study. As a pituitary hormone, GH is released into the peripheral blood through the pituitary portal vein and plays a vital role in mammary gland development and lactation. GH is a well-established treatment model used to study increased milk protein synthesis in bovine mammary tissues. A study on bovine mammary tissue suggested that increased milk protein synthesis occurs in response to GH treatment [39]. Some studies have also shown that GH acts indirectly on the bovine mammary gland by increasing blood flow and nutrient availability to the gland [40,41]. It is generally believed that GH acts directly or indirectly on mammary tissues through growth hormone receptors or IGF-1, respectively. Previously, researchers reported that circulating levels of IGF1 in plasma, ribosomal protein S6 (RPS6), and eukaryotic initiation factor 4E (eIF4E) abundance and phosphorylation in the mammary glands of lactating dairy cows increased in response to GH treatment [42,43], suggesting that GH may affect the synthesis of milk protein partly through IGF-1. Notably, our results, both in plasma hormones and proteomics, have shown that HMP cows always have higher IGF-1 concentrations than LMP cows. Additionally, KEGG analysis showed that the mTOR signaling pathway was enriched between the HMP and LMP groups, which is considered the key pathway regulating milk protein synthesis. The activation of mTOR is involved in initiating mRNA translation, resulting in the phosphorylation of S6K1, which is a downstream effector protein of mTOR.

Further in vitro studies on BMEC are warranted to better understand the molecular mechanism of IGF-1 in milk protein synthesis. We observed that the expression of β-casein increased in BMEC treated with IGF-1, and IGF-1 also activated the mTOR signaling pathway. It has been previously shown that IGF-1 acutely stimulated protein synthesis in BMECs [6], which is consistent with our results. It is important to note that a previous report stated that IGF1 signaling acting on BMEC did not occur through the autocrine pathway but rather through the endocrine pathway [39]. This indicates that the higher concentration of plasma IGF-1 in HMP cows was transported to the mammary gland, which enhanced milk protein synthesis via the mTOR signaling pathway.

## 5. Conclusions

In conclusion, the present study highlighted the differences in the plasma proteome, hormones, and biochemical indicators of Holstein cows with different long-term milk protein concentrations. Several hormones, proteins, and physiological pathways, including INS, GH, PRL, FN1, and ADIPOQ, particularly the IGF-1 and mTOR signaling pathways, potentially contribute to the phenotype of different milk protein concentrations. Moreover, we demonstrated that β-casein synthesis was enhanced via the mTOR signaling pathway following IGF-1 treatment using an in vitro model. In the bovine mammary gland, the coordination of these hormones and nutrient factors relies on a complex process that is yet to be fully elucidated. Thus, further research on identifying and understanding the molecular mechanisms is required to better understand how these factors are responsible for the differences in milk protein concentrations.

## Figures and Tables

**Figure 1 biology-11-01239-f001:**
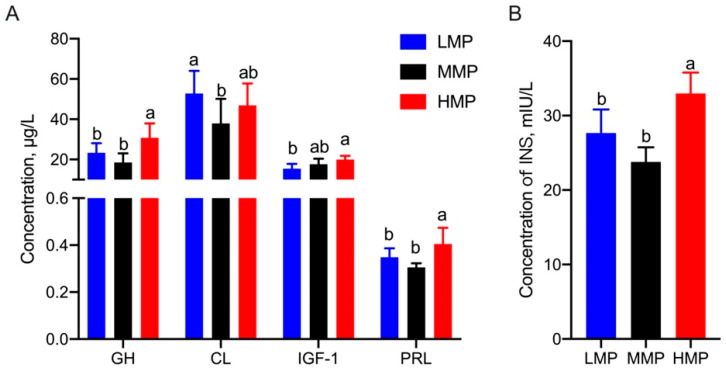
Differential hormones in plasma in dairy cows with different milk protein concentrations. (**A**) Concentrations of GH, CL, IGF-1 and PRL in plasma. (**B**) Concentration of INS in plasma. This experiment was repeated 3 times, and data are reported as mean ± SEM; different superscript letters in the same row indicate significant differences between groups (*p* < 0.05). LMP: low milk protein concentration group; MMP: medium milk protein concentration group; HMP: high milk protein concentration group.

**Figure 2 biology-11-01239-f002:**
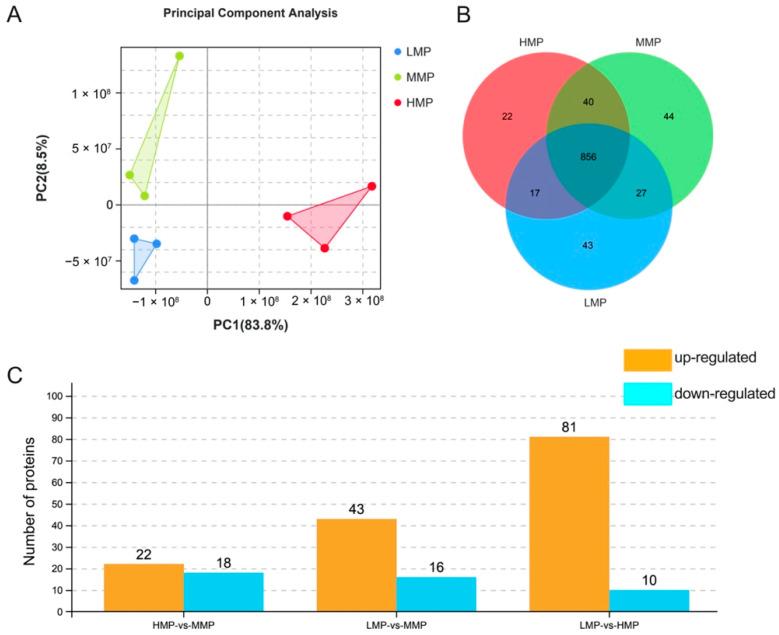
PCA and characteristics of plasma proteome among three groups. (**A**) PCA analysis of plasma. (**B**) Plasma proteins identified in three groups of cows. (**C**) Number of differential proteins identified in the plasma of three groups of cows. LMP: low milk protein concentration group; MMP: medium milk protein concentration group; HMP: high milk protein concentration group. The group presented before the symbol “vs” was a control group.

**Figure 3 biology-11-01239-f003:**
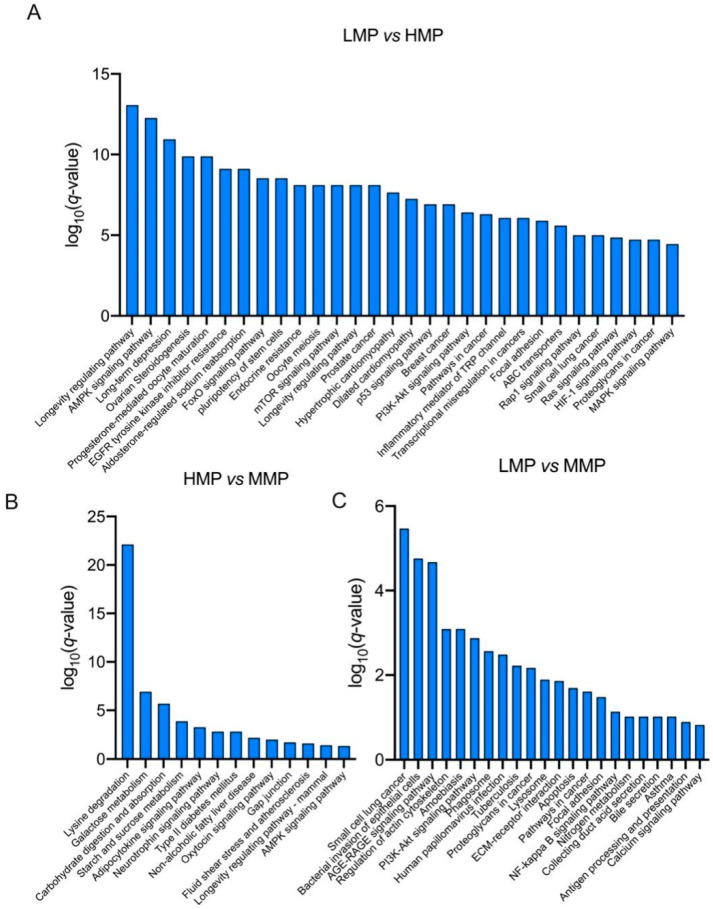
KEGG pathways enrichment of differential proteins in plasma between (**A**) LMP and HMP, (**B**) HMP and MMP, (**C**) LMP and MMP. LMP: low milk protein concentration group; MMP: medium milk protein concentration group; HMP: high milk protein concentration group. The group presented before the symbol “vs” was a control group.

**Figure 4 biology-11-01239-f004:**
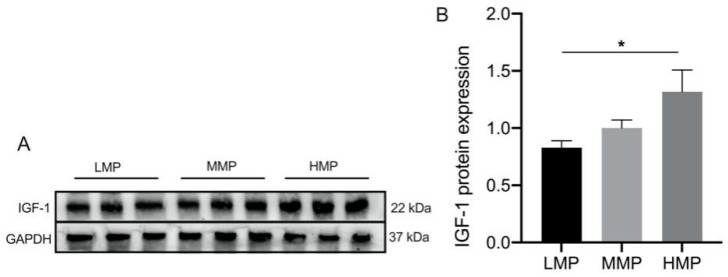
Expression of IGF-1 protein in plasma among three groups. (**A**) Representative Western blotting of IGF-1. (**B**) Protein abundance of the IGF-1. This experiment was repeated 3 times, and data are reported as mean ± SEM. * *p* < 0.05. LMP: low milk protein concentration group; MMP: medium milk protein concentration group; HMP: high milk protein concentration group.

**Figure 5 biology-11-01239-f005:**
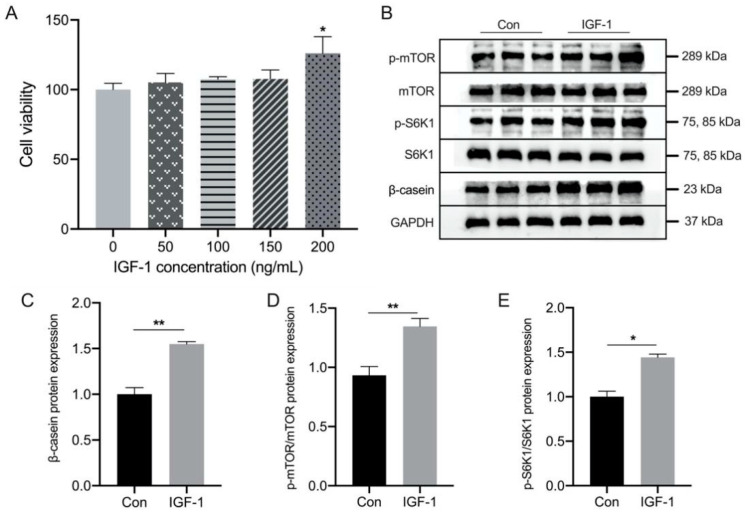
Effects of IGF-1 concentration on cell viability and protein expression of β-casein in bovine mammary epithelial cells (BMEC). (**A**) Relative cell viability after IGF-1 treatment. (**B**) Representative Western blotting of β-casein, p-S6K1, S6K1, p-mTOR, and mTOR. Protein abundance of the (**C**) β-casein, phosphorylation state, (**D**) p-S6K1/S6K1, and (**E**) p-mTOR/mTOR. This experiment was repeated 3 times, and data are reported as mean ± SEM. * *p* < 0.05 and ** *p* < 0.01. LMP: low milk protein concentration group; MMP: medium milk protein concentration group; HMP: high milk protein concentration group.

**Table 1 biology-11-01239-t001:** Plasma biochemical parameters of cows with different milk protein concentrations.

Parameter	Group			*p*-Value ^1^
	LMP	MMP	HMP	
Aspartate aminotransferase, IU/L	109.89 ± 7.12 ^a^	86.22 ± 5.78 ^b^	90.22 ± 4.63 ^b^	0.02
Alanine aminotransferase, IU/L	36.30 ± 3.33 ^a^	28.40 ± 2.42 ^b^	38.00 ± 2.00 ^a^	0.04
Total protein, mmol/L	69.34 ± 1.62 ^ab^	65.10 ± 2.45 ^b^	72.29 ± 2.80 ^a^	0.11
Albumin, g/L	29.15 ± 0.56	28.54 ± 0.86	28.36 ± 0.78	0.73
Globulin, g/L	40.19 ± 1.11 ^ab^	36.55 ± 1.90 ^b^	43.93 ± 2.17 ^a^	0.02
Total bilirubin, μmol/L	0.99 ± 0.11	1.65 ± 0.90	0.86 ± 0.20	0.57
Total bile acid, μmol/L	31.50 ± 7.88	24.68 ± 4.52	31.96 ± 3.86	0.61
Lactate dehydrogenase, g/L	955.50 ± 36.31	902.20 ± 50.71	939.44 ± 39.69	0.66
α-Hydroxybutyric acid, g/L	970.70 ± 43.15	978.20 ± 69.51	968.00 ± 48.34	0.99
Urea, mmol/L	5.04 ± 0.26	4.15 ± 0.29	4.36 ± 0.52	0.20
Creatinine, umol/L	61.05 ± 3.13	60.72 ± 5.77	65.49 ± 2.63	0.68
Glucose, mmol/L	3.16 ± 0.06 ^b^	3.27 ± 0.11 ^ab^	3.50 ±0.11 ^a^	0.06
Total cholesterol, mmol/L	5.10 ±0.34 ^a^	4.29 ± 0.36 ^ab^	3.80 ± 0.26 ^b^	0.03
Triglyceride, mmol/L	0.14 ± 0.01	0.18 ± 0.04	0.16 ± 0.01	0.52
High-density lipoprotein, mmol/L	1.70 ± 0.06 ^a^	1.35 ± 0.08 ^b^	1.43 ± 0.07 ^b^	<0.01
Low-density lipoprotein, mmol/L	1.94 ± 0.16 ^a^	1.61 ± 0.14 ^ab^	1.47 ± 0.11 ^b^	0.07

Note: LMP: low milk protein concentration group; MMP: medium milk protein concentration group; HMP: high milk protein concentration group. ^1^
*p*-value among three groups. Data were expressed as mean ± SEM; different superscript letters in the same row indicate significant differences between groups.

**Table 2 biology-11-01239-t002:** Differentially expressed protein in plasma (*p* < 0.05, FC > 1.5).

ID	Protein	Group					
		MMP vs. HMP		MMP vs. LMP		LMP vs. HMP	
		log_2_FC ^1^	*p*-Value ^2^	log_2_FC	*p*-Value	log_2_FC	*p*-Value
NP_783630.2	CGN1	1.090	0.003	0.189	0.031	0.900	0.000
NP_001029390.1	FETUB	−1.120	0.000	−1.563	0.000	0.444	0.000
XP_015327629.1	ADIPOQ	0.663	0.001	−0.454	0.000	1.117	0.000
NP_803450.2	TF	0.273	0.000	−0.349	0.693	0.622	0.000
NP_777246.1	SERPING1	0.207	0.077	−0.493	0.000	0.700	0.000
NP_001015590.2	ITIH4	0.456	0.078	−0.262	0.000	0.717	0.000
NP_001137569.1	CRP	0.007	0.127	1.181	0.000	−1.174	0.000
NP_001157250.1	FN1	0.354	0.602	−0.650	0.000	1.004	0.000
NP_001159957.1	C4A	0.320	0.513	−0.343	0.000	0.663	0.000
NP_001094702.1	CPN2	0.147	0.085	−0.673	0.001	0.820	0.000
NP_001193699.1	MAP2K5	0.627	0.002	−0.262	0.291	0.890	0.000
XP_005202014.2	ICA	0.005	0.390	−0.587	0.002	0.593	0.000
NP_001033643.1	ADPRH	0.416	0.043	−0.312	0.073	0.728	0.002
NP_001095368.1	ITIH3	−1.075	0.000	−0.533	0.010	−0.542	0.002
NP_001106748.1	KNG2	−0.572	0.030	−0.982	0.005	0.409	0.003
XP_002699259.1	MSANTD2	1.229	0.342	−0.425	0.664	1.654	0.003
NP_777253.1	GLYCAM1	0.794	0.055	0.095	0.744	0.699	0.004
XP_005228030.1	ATRX	1.079	0.300	0.264	0.757	0.815	0.004
NP_001073103.1	CRISP3	0.571	0.352	−0.569	0.006	1.140	0.005
NP_001069646.1	CFP	0.257	0.241	−0.507	0.430	0.764	0.010
NP_001094532.1	PF4	−0.230	0.872	−0.832	0.067	0.602	0.010
XP_024855781.1	Gucy1b2	0.558	0.137	−0.357	0.401	0.915	0.012
NP_001039988.1	HAVCR1	−0.026	0.168	0.576	0.088	−0.602	0.014
NP_001179392.1	GOLM1	1.777	0.184	0.285	0.779	1.493	0.016
NP_001107218.1	ABCA3	0.459	0.012	−0.300	0.242	0.759	0.020
NP_001033763.1	LBP	0.334	0.969	−0.687	0.008	1.021	0.022
XP_024838406.1	ITIH4	0.703	0.184	−0.324	0.693	1.026	0.023
NP_001033174.1	RPP38	−0.428	0.190	0.544	0.195	−0.972	0.024
XP_002689991.2	SUSD1	−0.431	0.581	0.432	0.256	−0.864	0.026
NP_001071296.1	IGF1	−0.148	0.529	−0.814	0.135	0.666	0.028
NP_001069769.1	APOD	0.467	0.078	0.785	0.023	−0.318	0.029
NP_776392.1	TTR	0.788	0.000	0.093	0.019	0.695	0.030
NP_001035678.1	SYCP3	0.332	0.152	−0.280	0.425	0.613	0.034
XP_005216584.2	SERPING1	1.627	0.056	0.393	0.037	1.234	0.045
XP_005217472.2	CFHR5	1.445	0.818	0.811	0.338	0.633	0.050
NP_001029638.1	APCS	1.150	0.000	0.756	0.392	0.394	0.056
XP_005217494.2	FGA	−0.363	0.050	−1.018	0.022	0.655	0.074
XP_010815084.2	C6	0.627	0.003	0.837	0.025	−0.210	0.117
NP_001180035.1	Gripap1	−0.354	0.473	1.698	0.040	−2.052	0.170
NP_001070299.1	COL3A1	1.742	0.040	−0.752	0.671	2.494	0.291
NP_001093176.1	ECM1	0.083	0.933	−0.785	0.047	0.867	0.299
NP_001029392.1	CCL4	−0.859	0.291	−0.606	0.047	−0.254	0.379
NP_777124.1	CHIA	−0.418	0.023	−0.855	0.017	0.437	0.496
NP_001073741.2	OSMR	−0.395	0.198	−0.737	0.000	0.342	0.533
NP_848667.1	CA2	−1.350	0.346	−1.767	0.039	0.417	0.563
NP_776532.1	MBL	−0.217	0.581	−0.692	0.016	0.475	0.605
NP_001029784.1	HPX	−0.345	0.125	−0.744	0.041	0.399	0.641
NP_776359.1	LUM	−0.324	0.079	−0.585	0.007	0.261	0.649
XP_024847454.1	FBLN1	−0.626	0.008	−0.963	0.016	0.337	0.668
XP_003587429.1	Wfdc18	−0.513	0.488	−0.637	0.049	0.124	0.679
XP_005205978.1	MGAM	0.896	0.000	0.552	0.011	0.344	0.692
NP_001028787.1	CTSS	−0.246	0.302	−0.779	0.000	0.532	0.702
XP_024837895.1	SERPINA3−7	−0.690	0.254	−0.817	0.004	0.126	0.770
NP_001002237.1	CL43	0.575	0.048	−0.666	0.016	1.241	0.778
XP_010808587.1	Itsn2	−0.489	0.051	−0.665	0.019	0.175	0.808
XP_024847033.1	KMT2C	−0.830	0.008	−0.777	0.062	−0.053	0.817
XP_024833616.1	IGLL5	−0.394	0.001	−1.325	0.000	0.931	0.930
NP_001192115.1	IGHA2	−0.648	0.219	−1.576	0.022	0.928	0.967

Note: LMP: low milk protein concentration group; MMP: medium milk protein concentration group; HMP: high milk protein concentration group. log_2_(FC) > 0 means that this protein is higher in the group after the “vs” than in the group before the “vs”; log2(FC) < 0 means that this protein is lower in the group after the “vs” than that in the group before the “vs”. ^1^ FC, fold-change for concentration of proteins. ^2^
*p*-value < 0.05 was regarded as statistically different.

## Data Availability

The data that support the findings of this study are available from the corresponding author upon reasonable request.

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
