# Peer review of "Plasma Proteomic Profiling Reveals the Regulatory Factors of Milk Protein Synthesis in Holstein Cows"

_biology, 2022, doi:10.3390/biology11081239_

Round 1
Reviewer 1 Report
Manuscript entitled “Plasma Proteomic Profiling Reveals the Regulatory Factors of Milk Protein Synthesis in Holstein Cows”. The article demonstrated the differences in plasma proteins of dairy cows differing different milk protein concentrations. The grouping basis (cows with different long-term milk protein concentrations) seems to be interesting, although the description of the methods is insufficient. Authors should consider the following suggestions:
1. It is not clear how many data points for all cow were used. Authors indicate there were no differences in parity or milk yield. Some procedure had to be used to make them similar without previously knowing the data (Blocking?)
2. What is the reasoning for sampling before morning feeding? Were they sampled just once? Were all 30 cows sampled on the same day? If not, how was sampling distributed among cow categories?
3. Correct the spelling of “μmol” in Table 1and Fig 1A, not “umol”.
4. Line 192 Give the concentrations and composition of the diluent of the primary antibodies.
5. The corresponding number of table 2 footnotes were listed repeatedly.
6. Figure 3 What is the basis for the author to select these pathways for showing.
7. It is not clear why the IGF-1 was used in further vitro experiment, rather than other differentially expressed proteins, please explain.
8. Line 354-355 Delete “As stated above, LMP cows had a lower plasma Glc concentration than HMP cows”, it is meaningless.
9. Line 361 “were” instead of “are”.
10. Line 93-94:Is the grouping of the 3 groups of cows (LMP, MMP, HMP) determined by collecting milk samples from the cows and testing their milk composition? If so, it is necessary to supplement the method and process of milk protein detection.
11. What do the SEM and P-values in Table 1 represent? Between the three groups? And why is there no corresponding SEM in single group?
Author Response
Response to Reviewer 1 Comments
Manuscript number:biology-1877993
Title: Plasma Proteomic Profiling Reveals the Regulatory Factors of Milk Protein Synthesis in Holstein Cows
Thank you very much for the reviewer’s evaluation and comments on our paper. We have revised the manuscript according to your kind advice and detailed suggestions. Enclosed please find the responses to the reviewer. We sincerely hope this manuscript will be finally acceptable to be published. Thank you very much for all your help.
Point 1:
It is not clear how many data points for all cow were used. Authors indicate there were no differences in parity or milk yield. Some procedure had to be used to make them similar without previously knowing the data (Blocking?)
Response 1:
Thank you for your comments. In the preliminary work, we collected the milk protein contents data from DHI data (10 data points, recorded once a month) provided by a commercial farm, and we have added the description in the revised manuscript (Line 109). The specific procedure has described in our previous study1, and the multivariate analysis of variance (ANOVA) using the General Linear Model procedure to guarantee parity, day in lactation and milk yield have no effect on milk protein concentrations of 30 experimental cows. The effect of parity, milk yield and days in lactation on milk protein concentration of the 30 cows is shown in the table.
|
Parameter |
Degree of freedom |
Sum of squares |
Mean square |
F-value |
P-value |
|
Parity |
1 |
0.001 |
0.001 |
0.009 |
0.928 |
|
Milk yield |
16 |
1.098 |
0.069 |
0.450 |
0.912 |
|
Days in lactation |
4 |
0.143 |
0.036 |
0.235 |
0.910 |
[1]. Wang, X. L.; Zeng, H. F.; Xu, J.; Zhai, Y. F.; Xia, H. B.; Xi, Y. M.; Han, Z. Y. Characteristics of ruminal microbiota and metabolome in Holstein cows differing in milk protein concentrations. Journal of Animal Science. 2022.
Point 2:
What is the reasoning for sampling before morning feeding? Were they sampled just once? Were all 30 cows sampled on the same day? If not, how was sampling distributed among cow categories?
Response 2:
Thank you for your comments. We refer to previous studies and collect samples before morning feeding [1]. Because some components in blood will influenced after feeding. Collecting samples before morning feeding can keep the indexes of dairy cows at the same level as much as possible. Cows were sampled on the same day and sampled just once, and we have added some descriptions in manuscript.(Line 125)
[1] Xue M Y, Sun H Z, Wu X H, et al. Assessment of rumen bacteria in dairy cows with varied milk protein yield[J]. Journal of Dairy Science, 2019.
Point 3:
Correct the spelling of “μmol” in Table 1and Fig 1A, not “umol”.
Response 3:
Thank you for your comment, and we have modified the spelling of “μmol” in Table 1and Fig 1A.
Point 4:
Line 192 Give the concentrations and composition of the diluent of the primary antibodies.
Response 4:
Thank you for your comment, and we have modified the descriptions.(Line 210-224)
Point 5:
The corresponding number of table 2 footnotes were listed repeatedly.
Response 5:
Thank you for your careful review, we have modified the table 2.
Point 6:
Figure 3 What is the basis for the author to select these pathways for showing.
Response 6:
Thank you for your comments, KEGG pathways showed in figure 3 were represent the pathways enrichment of all differential proteins among groups, and we described it in figure note. (Line 335)
Point 7:
It is not clear why the IGF-1 was used in further vitro experiment, rather than other differentially expressed proteins, please explain.
Response 7:
Thank you for your comments. Both results of plasma proteome and hormones showed that IGF-1 level was higher in HMP group than LMP group, so we speculated that IGF-1 may play a unique role in milk protein synthesis.
Point 8:
Line 354-355 Delete “As stated above, LMP cows had a lower plasma Glc concentration than HMP cows”, it is meaningless.
Response 8:
Thank you for your suggestion, we have deleted this sentence.
Point 9:
Line 361 “were” instead of “are”.
Response 9:
Thank you for your comments, and we have modified the description. (Line 417)
Point 10:
Line 93-94:Is the grouping of the 3 groups of cows (LMP, MMP, HMP) determined by collecting milk samples from the cows and testing their milk composition? If so, it is necessary to supplement the method and process of milk protein detection.
Response 10:
Thank you for your comments. The milk protein concentration data were obtained from the DHI data provided by the farm, and the DHI data was measured according to the standard process.
Point 11:
What do the SEM and P-values in Table 1 represent? Between the three groups? And why is there no corresponding SEM in single group?
Response 11:
Thank you for your valuable suggestion. The SEM and P-value in our original manuscript represent the SEM value and P-value among three groups. We have modified the Table 1 in manuscript as your suggestion.

Reviewer 2 Report
The artical is written well and has its significance in the field of dairy as it is focused on understanding the mechanism of Regulatory Factors of Milk Protein Synthesis in Holstein Cows. it will also help to identify the allergic proteins in cow milk.
Author Response
Response to Reviewer 2 Comments
Manuscript number:biology-1877993
Title: Plasma Proteomic Profiling Reveals the Regulatory Factors of Milk Protein Synthesis in Holstein Cows
Thank you for the reviewer’s valuable comments. Those comments are all valuable and very helpful for revising and improving our paper, as well as the important guiding significance to our researches. We would like to express our great appreciation to you for the serious and responsible reviews of the article. We have revised the manuscript accordingly and hope meet with your approval.

Author Response
Response to Reviewer 3 Comments
Manuscript number:biology-1877993
Title: Plasma Proteomic Profiling Reveals the Regulatory Factors of Milk Protein Synthesis in Holstein Cows
Thank you for the reviewer’s valuable comments. We really appreciate your efforts in reviewing our manuscript. We have revised the manuscript accordingly. We have studied comments carefully and have made correction which we hope meet with approval. Our point-by-point responses are detailed below.
Point 1:
Line 93-95: LMP, MMP, HMP please, provide the full name of the abbreviations of groups on first mention in the text
Response 1:
Thank you for your careful review, we have added the the full name of the abbreviations in manuscript.(Line 112-114)
Point 2:
Line 104: temperature is missing Line
Response 2:
Thank you for your careful review, we have added the the temperature.(Line 127)
Point 3:
Line 108-110: please could you briefly add some information about protocol?
Response 3:
Thank you for your comment, and we have modified the description here.(Line 132-134)
Point 4:
Line 113: “Function” is the manufacturer? I cannot find their website.
Response 4:
Thank you for your comment, and we have corrected the description. The manufacturer is Beijing Fangcheng Biotechnology Co., Ltd.(Line 138)
Point 5:
Line 113-118: Could you briefly add some information about protocol? What type of ELISA did you used? Competitive, sandwich? Which software did you used to interpolate the sample concentrations from the standard curve?
Response 5:
Thank you for your suggestion. We have added the description of “Hormone concentration Assay” part.(Line 137-138, 142)
Point 6:
Line 168: manufacturer of microplate reader is missing
Response 6:
Thank you for your suggestion, and we have added the manufacturer of microplate reader.(Line 199)
Point 7:
Line 172: reference is missing. please add some details about sample preparation
Response 7:
Thank you for your suggestion. We have added the reference and some descriptions here.(Line 202-205)
Point 8:
Line 177: What exactly were the secondary antibodies?
Response 8:
Thank you for your comments, the secondary antibodies used in our study was secondary antibodies conjugated to HRP goat anti-rabbit IgG. We have corrected it in the revised manuscript. (Line 210-211)
Point 9:
Line 184: Which software did you used to quantify the band intensities?
Response 9:
Thank you for you comment. Image J 1.8.0 software was used to quantify the densitometry readings of the bands, and we have added descriptions here.(Line 220-221)
Point 10:
Line 186: The PCA and KEGG analysis are not included in the chapter of Material and Methods. Maybe the name of “Statistical analysis” could be changed to “Data analysis” and you can also add some information about PCA and KEGG.
Response 10:
Thank you for your suggestion, and we have added the information about PCA and KEGG analysis in Line 184-186.
Point 11:
Table 1: change item to parameter Table 1: The P values of Total protein, Glucose, Low density lipoprotein are higher than 0.05, but they are indicated as significant. The table caption contains P<0.05. Table 1: It would be easier to interpret if you also use an asterix (*) next P value, where you detect significant differences (of course the letters must remain).
Response 11:
Thank you for your comments. We have corrected the “Item” to “Parameter” in Table 1. P-value in table 2 is represent the P-value among three groups, not between every two groups, and we have added the table footnote in manuscript to make the description clear as your suggestion.(Line 255-264)
Point 12:
Figure 1: It might be useful if the P values of hormones were included in the figure or text.
Response 12:
Thank you for your suggestion, we have added the descriptions of P-value in text.(Line 248-254)
Point 13:
Line 225: DIA technology was performed to identify plasma protein differencies
Response 13:
Thank you for your suggestion, we have modified this sentence.(Line 273)
Point 14:
Line 230-233: The text and Figure 2C are different. In my opinion the figure is correct. The text should be modified. An example of the correction: The results showed that the MMP group compared to HMP group had 22 upregulated and 18 downregulated proteins, and 43 upregulated, and 16 downregulated proteins were observed in the MMP compared to LMP group, respectively (P < 0.05). 81 upregulated and 10 downregulated proteins were identified in the HMP group compared with the LMP group (P < 0.05) (Figure 2C).
Response 14:
Thank you for your valuable comments, we have modified the text as your suggestion.(Line 289-291)
Point 15:
Figure 2C: The y axis is missing the unit.
Response 15:
Thank you for your careful review, and the Y axis was added in Figure 2C.
Point 16:
Table 2: Why is the term “metabolite” used in the caption instead of protein?
Response 16:
Thank you for your comments, we have corrected the “metabolite” to “protein”.(Line 326-329)
Point 17:
Figure 5: In addition to the specific bands of p-mTOR and p-S6K1, non-specific bands can be seen on the blots. Could be due to incomplete blocking or non-specific binding of antibodies? This phenomenon can reduce the accuracy of the results.
Response 17:
Thank you for your comments. A possible reason for the appearance of non-specific bands of p-mTOR and p-S6K1 was that the primary antibodies we used here were derived from human (had cross reaction with cells of bovine) due to the scarcity of primary antibodies derived from bovine. However, the WB process was performed in strict accordance with the manufacturer's instructions, and the target protein was distinguished according to its molecular weight.
Point 18:
Line 298: 91 differentially expressed proteins (in abstract line 17)
Response 18:
Thank you for you comment, we have modified the description.(Line 371)

This manuscript is a resubmission of an earlier submission. The following is a list of the peer review reports and author responses from that submission.